# Letting Go as an Aspect of Rumination and Its Relationship to Mindfulness, Dysphoria, Anxiety, and Eudemonic Well-Being

**DOI:** 10.3390/bs12100369

**Published:** 2022-09-29

**Authors:** Jesse R. Caswell, Aishwarya N. Duggirala, Paul Verhaeghen

**Affiliations:** School of Psychology, Georgia Institute of Technology, 654 Cherry Street NW, Atlanta, GA 30332, USA

**Keywords:** rumination, anxiety, depression, letting go, mindfulness

## Abstract

**Background:** We examined how a newly proposed facet of rumination, that is, the (in)ability to let go, might relate to other aspects of rumination and to psychological outcomes. **Methods:** In two independent samples (*n* = 423 and 329, resp.) of college students, we measured a broad set of rumination and rumination-related measures, letting go, anxiety and dysphoria; in the second sample, we also collected data on mindfulness, self-compassion and eudemonic well-being. **Results:** Factor analysis of rumination and rumination-related measures yielded three factors: (a) negative intrusive thought; (b) reflectiveness; and (c) the inability to let go. Repetitive intrusive thought and the ability to let go were significant (and thus partially independent) predictors for the three outcomes of anxiety, dysphoria, and wellbeing. The inability to let go and repetitive intrusive thought significantly mediated between mindfulness and all three outcomes. **Conclusions:** The findings suggest that letting go is a potentially interesting aspect of rumination not fully captured in the traditional concept of rumination and its standard measures.

## 1. Introduction

Rumination can be defined as repetitive, prolonged, and recurrent negative thinking about oneself, one’s emotions, and one’s (upsetting) experiences (e.g. [1]). Such thinking is a risk factor for both the onset and continued maintenance of both depression (e.g., [2,3]) and anxiety (e.g., [4]). It is often perceived as intrusive and difficult to disengage from [5].

Many mechanisms—not mutually exclusive—for the link between rumination and negative psychological outcomes have been proposed. Rumination, for instance, serves to exacerbate and prolong existing negative emotional states (e.g., [6]). It also interferes with positive coping behaviors, such as problem solving (e.g., [7]) and active instrumental behavior (e.g., [8]). We will refer to this as maladaptive rumination. In contrast, rumination can also have positive consequences by allowing for *reflection* on the past in the service of adaptive preparation for the future [9]. We will refer to this as adaptive rumination.

One often-proposed underlying cognitive mechanism for maladaptive repetitive rumination is a lack of executive control over the contents of awareness. This lack of control applies maybe especially when awareness is awash with negativity, which in turn leads to the continued cycling and likely exacerbation of negative emotions and cognitions (e.g., [10])—and unneeded and probably unwanted stickiness of thought. In line with this claim, a recent meta-analysis [11] found that repetitive negative thinking (typically measured by standard rumination measures, such as the Ruminative Response Scale [RRS] and its Brooding subscale [12]) was reliably (*r* = −0.20) associated with one specific aspect of executive control, namely the ability to discard no longer relevant material from working memory. 

In the literature that intersects the study of rumination and that of mindfulness, this particular ability has been labeled ‘*letting go*’ [13,14]. Frewen et al. claimed that it is exactly this ability that underlies the beneficial effects of trait mindfulness or of mindfulness interventions on rumination, which in turn leads to lower levels of dysphoria or anxiety. The idea is that a mindful attitude, that is, a non-judgmental, non-conceptual, and accepting form of awareness of one’s mental, emotional, and bodily-sensory experiences (e.g., [15]), allows one to view negative thoughts as temporary events that may capture attention, but do not necessitate a reaction. Thus, instead of identifying with the thoughts or trying to suppress them, one can simply accept, decenter, and let go. This letting-go attitude “cultivate[s] one’s inner capacity to reflect upon and influence one’s own cognitive experiences. This purposeful orientation toward one’s thoughts may promote affect regulation through cognitive flexibility” [13]. In other words, letting go should diminish maladaptive rumination and promote reflection, both of which would be beneficial to mood and wellbeing. 

The ability to let go can obviously be measured using objective measures of cognitive control [11], but the nomothetic span and ecological validity of these remain unknown. A more targeted self-report measure has been proposed [13], in which participants indicate the degree to which they are able to let go of negative automatic thoughts in specified life or symptom domains. Data from this inventory, then, can give us some indication of how individuals perceive their own propensity to let go, and provide some valuable insight in how this propensity connects to habitual mindfulness (as its potential source) on the one hand and rumination, depression, and anxiety (as possible outcomes) on the other. With the regard to the first part of the equation, Frewen et al. obtained (mostly) significant correlations between the ability to let go and measures of trait mindfulness (*r*s between 0.19 and 0.43); the ability to let go was also substantially boosted after a mindfulness intervention, suggesting a causal link. Data on the second part, however, have, as far as we know, not been collected. This also does not allow us to examine the logical next step, namely, to check whether letting go is a significant mediator between mindfulness on the one hand and rumination, anxiety, and depression on the other. This, then, is what we set out to do in our study. 

We investigated the link between rumination, letting go, anxiety and depression in two large and independent samples of undergraduate students; in the second of these samples, we also collected data on mindfulness and eudemonic well-being. Note that we included a measure of personality, notably the Big Five, as a control measure, to partial out any influence of personality as a background measure. Such background or third-variable relationships are to be expected. For instance, neuroticism has often been found to correlate with both rumination and depression [16,17] as well as with mindfulness [18]; consequently, correlations between these three variables might well be due solely or in part to their underlying relationship with this personality trait. Likewise, conscientiousness is positively related to mindfulness [18] and negatively related to anxiety and depression (e.g., [19]) as well as rumination (e.g., [20]), creating the potential of background correlations between these variables as well.

In our analyses, we first conducted a factor analysis to examine whether letting go and rumination (operationalized under both its maladaptive negative repetitive thinking and its adaptive reflection aspect) are distinguishable constructs, as claimed by Frewen and colleagues [13]. 

Second, we investigated the relationship of these constructs (if distinguishable) to anxiety and depression (both operationalized here as continuous variables; we will therefore use the term dysphoria rather than depression in the remainder of this paper). We expected the inability to let go and repetitive negative thinking to be positively correlated to anxiety and dysphoria; the question of whether letting go adds additional variance to the prediction after the effects of negative repetitive thinking are taken into account is an open question. In the second sample, we used eudemonic well-being as an additional dependent variable, to examine whether the ability to let go and reflection would be positively related to positive psychological outcomes. Well-being is often operationalized as either hedonic (i.e., with a focus on pleasure or happiness—feeling good) or eudemonic (i.e., flourishing or actualization) [21]. We chose the latter as our outcome measure because it is more multidimensional and less directly related to our other outcomes, dysphoria and anxiety; eudemonic wellbeing is also more stable over time and has been found to be the driver of hedonic well-being rather than the other way around [22]. Previous work (e.g., [23]) has shown a modest connection between reflection and eudemonic well-being and a stronger connection with maladaptive rumination; the question whether letting go would be a predictor is still open. 

Third, in the second sample, we explored whether letting go is a mediator between mindfulness and these outcomes. Note that we implemented the concept of mindfulness here in a broader sense than is typically done. That is, usually mindfulness is defined as a particular (viz., open, non-judgmental, accepting) attitude or quality of attention—a specific form of self-awareness. Recent theoretical work, however, has extended this concept to also include self-regulation and self-transcendence [24], thus creating a mindfulness manifold. Using factor-analysis on two independent samples, we [25] found evidence that self-awareness has a more active, reflective aspect (which we labeled reflective self-awareness) as well as a more passive, non-judging aspect (which we labeled controlled sense-of-self in the moment), and that self-regulation could be fruitfully subdivided into self-preoccupation (a concept close to rumination) and self-compassion. In our own work using this expanded definition [25,26,27,28], we found that this expansion was helpful in further elucidating the effects of what is traditionally understood as mindfulness on outcomes as diverse as depression, stress, anxiety, wisdom, moral attitudes, prejudice, and compassion. More specifically, self-regulation and self-transcendence often added additional explanatory variance to the outcomes once self-awareness was taken into account, indicating their usefulness within both a clinical and positive psychology context. Note that because some of the aspects measured in our lab’s previous studies seem semantically quite close to (if not synonymous with) some of the rumination concepts measured here (viz., reflective self-awareness and reflection, and self-preoccupation and repetitive negative thinking), we operationalized the self-awareness aspect of mindfulness as the scores on one of the most often used mindfulness surveys, the Five Facets Mindfulness Questionnaire (FFMQ [29]), and simply dropped self-preoccupation from the analyses.

## 2. Methods

### 2.1. Participants

We collected data from two independent samples, both consisting of students at the Georgia Institute of Technology, collected independently over the course of the academic years 2020–2021 and 2021–2022. Note that all data collection occurred online due to the ongoing COVID-19 pandemic. Sample 1 consisted of 423 participants, 49.6% female, with a mean age of 19.7 (*SD* = 1.6). Sample 2 consisted of 329 participants, 49.2% female, with a mean age of 19.8 (*SD* = 1.8).

### 2.2. Measures

Note that Cronach alpha values reported here are those obtained for the present Sample 1 and Sample 2, respectively.

#### 2.2.1. The Big Five

We included the Mini-International Personality Item Pool (IPIP; [30]) as a 20-item measurement of the Big Five personality factors: Extraversion (Sample item: “I am the life of the party.”; alpha = 0.78 and 0.83, resp.), Agreeableness (sample item: “I sympathize with others’ feelings.”; alpha = 0.74 and 0.77, resp.), Conscientiousness (sample item: “I get chores done right away.”; alpha = 0.69 and 0.67, resp.), Openness (which the IPIP labels Intellect/Imagination; sample item: “I have a vivid imagination.”; alpha = 0.72 and 0.71, resp.), and Neuroticism (sample item: “I have frequent mood swings.”; alpha = 0.69 and 0.70, resp.), to control for the possible background correlations of our set of measures with personality in our multiple regression analyses.

#### 2.2.2. Rumination-Related Variables

**Rumination.** The Rumination-Reflection Questionnaire (RRQ [31]) includes two subscales: Reflection (12 items; sample item: “I love exploring my ‘inner’ self’”; alpha = 0.89 and 0.90, resp.), and Rumination (12 items; sample item: “Often I’m playing back over in my mind how I acted in a past situation.”; alpha = 0.91 and 0.92, resp.). The Broad Rumination Scale (BRS [32]) is a 29-item scale aimed at measuring ruminative behavior in a broad sense. Subscales are Optimism (five items; sample item: “My thoughts about myself are more often positive than negative.”; alpha = 0.78 and 0.80, resp.), Compulsivity (five items; sample item: “When I start to worry, it’s very hard for me to stop.”; alpha = 0.78 and 0.79, resp.), Social Expressiveness (two items; sample item: the reverse of “I do not like sharing my thoughts and feelings with others.”; alpha = 0.77 and 0.82, resp.), Broodiness (five items; sample item: “When something goes wrong, I tend to think of all the things that have recently gone wrong.”; alpha = 0.76 and 0.77, resp.), Distractibility (five items; sample item: “When I am emotional, it is hard for me to focus on what I am supposed to be doing.”; alpha = 0.75 and 0.79, resp.), Worrying (three items; sample item: “Uncertainty about the future bothers me.”; alpha = 0.51 and 0.70, resp.), and Reflectiveness (four items; sample item: “It is important for me to understand why I feel a certain way.”; alpha = 0.70 and 0.77, resp.).

**Intrusion and suppression.** The White Bear Suppression Inventory (WBSI [33]) is a 15-item questionnaire designed to measure thought intrusion (sample item: “My thoughts frequently return to one idea”; alpha = 0.82 and 0.84, resp.) and suppression (sample item: “There are things I prefer not to think about”; alpha = 0.79 and 0.79, resp.).

**Inability to let go.** The University of British Columbia Cognition Inventory Letting Go scale (UBC-LG [13]) measures the extent to which participants can let go of negative automatic thinking in particular domains. Here, we used four subscales: Depression (18 items; sample item: “I wish it would all end”; alpha = 0.95 and 0.95, resp.), Worry (eight items; sample item: “I’m afraid some harm will come to my friends”; alpha = 0.92 and 0.92, resp.), Social Fears (15 items; sample item: “I am going to be embarrassed”; alpha = 0.94 and 0.94, resp.), and Personal, where participants provide five personal thoughts that are worrisome and indicate how difficult it is to let these go (alpha = 0.84 and 0.84, resp.).

#### 2.2.3. Psychological Outcomes

**Dysphoria.** The Center for Epidemiological Studies-Depression (CES-D [6]; alpha = 0.94 and 0.94, resp.) is a 20-item self-report scale designed to measure self-reported symptoms associated with depression experienced in the past week. The items of the scale reflect six major facets of depression: depressed mood, feelings of guilt and worthlessness, feelings of helplessness and hopelessness, psychomotor retardation, loss of appetite, and sleep disturbance. 

**Anxiety.** The State-Trait Anxiety Inventory (STAI [34]) is a commonly used measure of trait and state anxiety that can be used in clinical settings to diagnose anxiety and to distinguish it from depressive syndromes. Form Y, its most popular version, includes two subscales, state anxiety and trait anxiety, Here, we used the state anxiety subscale (20 items; sample items: “I am tense; I am worried” and the reverse of “I feel calm; I feel secure.”; alpha = 0.93 and 0.94, resp.).

**Psychological Well-Being (only included in Sample 2).** Four subscales of the Psychological Well-Being Scale (PWB [35]) were included: Personal Growth (seven items; sample item: “I think it is important to have new experiences that challenge how you think about yourself and the world.”; alpha = 0.79), Positive Relations (seven items; sample item: “Most people see me as loving and affectionate.”; alpha = 0.75), Purpose in Life (seven items; sample item: “I have a sense of direction and purpose in life.”; alpha = 0.72), and Self-Acceptance (seven items; sample item: “When I look at the story of my life, I’m pleased with how things have turned out.”; alpha = 0.86). 

#### 2.2.4. The Mindfulness Manifold (Only Collected in Sample 2)

**Self-awareness.** The Five Facets Mindfulness Questionnaire (FFMQ [29]) is a 39-item questionnaire designed to measure five facets of mindfulness: Observing (sample item: “When I am walking, I deliberately notice the sensations of my body moving”; alpha = 0.74), Describing (sample item: “I am good at finding words to describe my feelings”; alpha = 0.87), Acting with awareness (sample item: the reverse of “When I am doing things, my mind wanders off and I am easily distracted”; alpha = 0.87), Nonjudging of inner experience (sample item: the reverse of “I criticize myself for having irrational or inappropriate emotions”; alpha = 0.85) and Nonreactivity (sample item: “I perceive my feelings and emotions without having to react to them”; alpha = 0.79).

**Self-compassion.** The Self-Compassion Scale, Short Form (SCS [36]), consists of 12 items, subdivided into six subscales of two items each: Self-Kindness (sample item: “I try to be understanding and patient towards those aspects of my personality I don’t like.”), Self-Judgment (sample item: “I am disapproving and judgmental about my own flaws and inadequacies.”), Common Humanity (sample item: “I try to see my failings as part of the human condition”), Isolation (sample item: “When I am feeling down, I tend to feel like most other people are probably happier than I am”), Mindfulness (sample item: “When something painful happens, I try to take a balanced view of the situation”), and Over-Identified (sample item: “When I fail at something important to me, I become consumed by feelings of inadequacy”). Here, we used the total score on all scales, reverse-coding when appropriate, so that higher scores indicate higher levels of self-compassion (alpha = 0.83) 

**Self-transcendence.** As in Verhaeghen (2019), self-transcendence was measured using a unit-weighted z-score composite of six scales. The Decentering subscale of the Experiences Questionnaire (EQ [37]; alpha = 0.86) consist of 13 items, measuring “the ability to observe one’s thoughts and feelings as temporary, objective events in the mind, as opposed to reflections of the self that are necessarily true”; sample item: “I am better able to accept myself as I am.” Four subscales of the Dispositional Positive Emotion Scales (DPES [38]) were included: Joy (six items; sample item: “I am an intensely cheerful person”; alpha = 0.81), Love (six items; sample item: “I develop strong feelings of closeness to people easily”; alpha = 0.79), Compassion (five items; sample item: “Taking care of others gives me a warm feeling inside.”; alpha = 0.86), and Awe (six items; sample item: “I see beauty all around me.”; alpha = 0.79) One subscale of the Aspects of Spirituality Scale (AS [39]) was included, namely, the Search for Insight/Wisdom Scale (seven items; sample item: “I strive for insight and truth”; alpha = 0.85). 

## 3. Results

### 3.1. Factor Analysis on the Rumination-Related Measures

Two exploratory factor analyses (principal component analysis with oblimin rotation), one for each sample, were conducted on the rumination-related scales (i.e., the RRQ, the BRS, the WBSI, and the UBC-LG scales). Scale or subscale scores (i.e., not item scores) were the unit of analysis. Eigenvalues and the scree plot suggested a 3-factor solution in Sample 1 and a 4-factor solution in Sample 2. In the latter case, the fourth factor was comprised of a single scale (BRS Social Expressiveness). The factor solution for each sample is presented in Table 1; the first three factors explain 60% of the variance in Sample 1, and 62% of the variance in Sample 2. We considered loadings >0.5 as interpretatively significant. The three first factors were largely identical across both samples. The first factor contained the maladaptive rumination scales of the RRQ and the BRS as well as both WBSI scales and the BRS Compulsivity subscale, suggesting a combination of maladaptive rumination and intrusive thought. We labeled this ‘repetitive intrusive thought’. A second factor was comprised of the two reflection/reflectiveness subscales (one from the RRQ, one from the BRS); we labeled this ‘reflectiveness’. A third factor included three of the four UBC-LG scales; we labeled this ‘inability to let go’. The two samples differ in some details—in Sample 1, UBC-LG Personal loaded on the first factor, in Sample 2, it loaded on the third; BRS Optimism loaded on the first factor in Sample 1, but not Sample 2. 

In practice, then, the factor solutions in the two samples largely converged. Consequently, we built unit-weighted *z*-score composites to denote the three aspects of rumination represented in our data set using the scales that converged onto identical factors across both data sets: (a) repetitive intrusive thought (unit-weighted *z*-score composite of RRQ Rumination, WBSI Intrusion, WBSI Suppression, BRS Compulsivity, BRS Worry, BRS Distractibility, and BRS Brooding); (b) reflectiveness (unit-weighted *z*-score composite of BRS Reflectiveness and RRQ Reflection), and (c) inability to let go (unit-weighted *z*-score composite of UBC-LG Social fear, UBC-LG Worry, and UBC-LG Depression).

### 3.2. Multiple Regression Analyses Predicting Dysphoria, Anxiety, and Well-Being from Repetitive Negative Thinking, Reflectiveness, and the (in)Ability to Let Go

Correlations between all relevant variables are reported in Table 2 and Table 3. Table 4 provides the results of regression analyses (one for each sample) predicting dysphoria and anxiety from the three rumination-related variables in both samples, as well as the analysis predicting well-being in Sample 2 from the three rumination-related variables. These analyses were set up as hierarchical regressions. In step 1, the background variables (gender and the Big Five) were entered. Step 2 adds the rumination-related variables, allowing us to examine their relationship to dysphoria and anxiety over and beyond that of the background variables. Step 3 added dysphoria as a predictor of anxiety and anxiety as a predictor for dysphoria and both as a predictor for well-being, to examine comorbidity and its potential mediating role.

Neuroticism and conscientiousness consistently predicted dysphoria, anxiety, and well-being, with neuroticism having a deleterious influence and conscientiousness having a beneficial influence (step 1). Repetitive intrusive thought and the ability to let go were significant (and thus independent) predictors for all three outcomes in step 2. Step 3 revealed that the influence of the inability to let go on anxiety was mediated through dysphoria, as shown by its now *ns* beta coefficient. For well-being, the effect of the inability to let go was likewise mediated through dysphoria, anxiety, or both. Step 3 also showed that repetitive intrusive thought had an independent influence on all three outcomes over and beyond the comorbidity between anxiety and dysphoria and the influence of anxiety and dysphoria on well-being. Note that multicollinearity was not a problem in these analyses; largest VIF = 2.38.

Table 5 presents results from Sobel tests to examine whether the three rumination-related variables are mediators between the seven aspects of mindfulness measured here and the three outcomes (Sample 2 only). Both the inability to let go and repetitive intrusive thought were significant mediators for all three outcomes. Reflectiveness was a less powerful mediator—it was not a mediator at all for anxiety, and it only mediated variance from observing and describing (the two more active forms of mindfulness) and self-compassion and self-transcendence to dysphoria and well-being.

### 3.3. Multiple Regression Analyses Predicting Repetitive Negative Thinking, Reflectiveness, and the (in)Ability to Let Go

Table 6 presents the result of a set of regression analyses (Sample 2 only) predicting the three rumination-related variables from the background variables and mindfulness. Repetitive intrusive thought was well predicted by self-awareness (four out of five subscales of the FFMQ) and (lack of) self-compassion; reflectiveness was positively related to self-compassion; and the inability to let go was related to non-judging and acting with awareness. Note that multicollinearity was not a problem in these analyses; largest VIF = 2.09. For the sake of completeness, Table 7 reports multiple-regression results predicting dysphoria, anxiety, and well-being from the full set of variables (gender, the Big Five, mindfulness, and the rumination-related variables).

## 4. Discussion

In this paper, we examined a possible facet of rumination, that is, the (in)ability to let go, and how this (in)ability might relate to anxiety and dysphoria in two independent samples of college students. Given that letting go, as a construct, was originally proposed from a mindfulness framework, we additionally examined, in our second sample, whether letting go would be a significant mediator between mindfulness on the one hand and rumination, anxiety, and depression on the other as claimed, but not verified, in that literature [13]. For good measure, we added eudemonic well-being, as a positive-psychology outcome, to the surveys in our second sample.

The two main results from the study are (a) that letting go can indeed plausibly be considered an aspect of rumination not fully captured in the standard way of measuring the concept, and (b) that the ability to let go indeed mediates between mindfulness and the psychological outcomes considered here. 

### 4.1. Letting Go as an Aspect of Rumination

Our findings suggest that letting go is a potentially interesting (and overlooked) aspect of rumination. They also suggest that this ability is not fully captured in the traditional concept of rumination, the traditional measures of the concept, or both. We highlight four pieces of evidence for this claim.

First, in the factor analysis of our set of rumination-related measures, where we purposefully cast a wide net, letting go emerged as a factor separate from both the more traditional measures of maladaptive rumination and reflectiveness. (Note that there was a discrepancy in the factor analyses, such that only the first three factors were identical across the two samples; Sample 1 additionally yielded a fourth factor consisting of a single scale (Social Expressiveness). Given that both samples are drawn from the same population, given that single-item factors are hard to interpret, given that Social Expressiveness per se does not function in our theoretical framework, and given that Social Expressiveness did not load on any factor in Sample 2, we decided to move forward with the three factors both samples had in common.) Interestingly, our measures of intrusion and suppression (both from the WBSI) loaded together with the more traditional rumination measures (notably RRQ rumination), whereas the letting go measures did not. This suggests that the traditional rumination concept captures the intrusive aspects of rumination quite well (which is why we labeled the corresponding factor repetitive intrusive thought), but fails to incorporate the inability to let go. The correlations between the composites representing the two factors, however, varied between samples (*r* = 0.08 for Sample 1 and 0.47 for Sample 2). This makes it hard to interpret its actual status in relation to these traditional measures–the Sample 1 data suggest independence, the Sample 2 data suggest a moderately strong relationship. The regression analyses on Sample 2 additionally suggest that the two factors predict each other even after gender, the Big Five, and mindfulness have been taken into account. 

Second, the inability to let go was a predictor of dysphoria, anxiety, and well-being over and beyond repetitive intrusive thought (i.e., what is typically understood as rumination; Table 4, step 2), demonstrating (a) that it is an important part of the puzzle in understanding mood disorders and well-being, and (b) that its influence is partially independent of that of repetitive intrusive thought. Step 3 in the same analyses gives us one possible idea of mechanism: Its influence on wellbeing and anxiety becomes non-significant, suggesting that the inability to let go might lead to increased dysphoria, which in turn might increase anxiety and decrease well-being. Longitudinal or experimental work would obviously be necessary to confirm this cascade of influences.

Third, the partial independence between letting go and repetitive intrusive thought is further illustrated by the differential ties of the two constructs to personality: Repetitive intrusive thought is strongly related to neuroticism, while the inability to let go is related to extraversion (Sample 1) and openness to experience (Sample 2).

Fourth, the inability to let go and repetitive intrusive thinking also have differential predictability from mindfulness (i.e., the FFMQ scales, self-compassion, and self-transcendence). In the regression analyses, repetitive intrusive thinking was predicted by (most of) the self-awareness aspects of mindfulness (viz., all subscales of the FFMQ, with the exception of describing), as well as by self-compassion; the proportion of variance explained was quite large (*R*^2^ = 0.60). Somewhat in contrast, the inability to let go was related to only two aspects of mindfulness, namely non-judging and acting with awareness, and its predictability was appreciably lower (*R*^2^ = 0.22). (We note that the inability to let go was significantly correlated with six out of seven aspects of mindfulness (thus replicating the findings of Frewen and colleagues [13]), but that the regression analysis revealed that some of these correlations are due to background correlations with the Big Five.) These regression findings reassert (e.g., [40]) that the self-awareness aspect of mindfulness is indeed a viable target if one wishes to affect dysphoria and anxiety through repetitive intrusive thought. Whether such interventions would also have much effect on the inability to let go remains to be seen, but the regression analyses suggest this would be less likely. 

### 4.2. Letting Go as a Mediator

As mentioned above, one key result is that the inability to let go is a significant predictor of dysphoria, anxiety, and well-being over and beyond repetitive intrusive thought, thus demonstrating that it is an important additional determinant of these psychological outcomes. It is also a significant mediator between mindfulness and these outcomes. A first nuance, however, is that the inability to let go appears to be a less powerful predictor than repetitive intrusive thought, in that the standardized coefficients associated with the inability to let go are generally at least nominally smaller than those associated with repetitive intrusive thought. Another is that in the case of anxiety and well-being, its influence was mediated through the more traditional aspect of repetitive intrusive thought. This suggests that the inability to let go contributes to intrusive thought, and it is this intrusiveness that, in turn, increases anxiety and decreases well-being. Again, longitudinal or experimental research would be useful to help determine the temporal or causal sequencing of these processes.

It remains an open question as to what underlies the ability to let go. Our analyses suggest that it has few determinants in either personality or mindfulness. One possibility is the claim entertained in the Introduction, namely that the ability to let go is in essence a cognitive variable, that is, an individual’s estimation of their capability for executive control. This possibility remains to be investigated.

### 4.3. Additional Results

One additional result concerns the role of reflectiveness. First, reflectiveness turned out to be a less powerful mediator than the two other aspects of rumination—it is not a mediator at all for anxiety, and it only mediates variance from observing, describing, self-compassion, and self-transcendence to dysphoria and well-being. The self-awareness mediators do make sense– observing and describing are the two more active, reflective forms of mindfulness. Second, reflectiveness is itself predicted mostly by openness to experience and self-compassion. We believe that the role of self-compassion as found here is new and worthy of further investigation. One possibility is that self-compassion–an open attitude in which one accepts one’s humanity and one’s flaws with kindness–is a condition for reflectiveness to become truly possible. That is, self-compassion might create a form of inner safe space that allows for critical self-examination. Conversely, the causal arrow might point in the other direction, namely that open self-examination allows one to realize that self-directed kindness and compassion is in order. Apart from its positive relationship to reflectiveness, self-compassion also negatively predicted repetitive intrusive thought. This in turn suggests that self-compassion in and of itself might be an interesting fulcrum for interventions aimed at diminishing dysphoria and anxiety and promoting flourishing (e.g., [41]).

Self-transcendence, the final aspect of mindfulness considered here, was a strong predictor of all three outcomes after all other variables were taken into account. It did not, however, predict any of the three rumination-related variables, suggesting that its influence is either direct or via an unidentified third variable. The beneficial role of self-transcendence is hardly surprising, given that this variable has been found to be related to a large number of beneficial outcomes, including wisdom, moral attitudes, prejudice, and compassion [25,26,27,28].

Finally, we should highlight the role of personality. As in our previous work [25,26,27,28], we found a few instances where the Big Five led to spurious correlations between variables of interest, that is, correlations that are due to the background relationship of both variables with one or more aspects of the Big Five. The inability to let go was significantly correlated with six out of seven aspects of mindfulness, but only two of those survived multiple regression. The likely culprit here is the background correlations with neuroticism. This finding goes against the theoretical framework [13,14] that claims that letting go is a direct effluent of mindfulness. Likewise, the significant correlations between reflectiveness and the mindfulness variables of observing, describing, and self-transcendence did not survive multiple regression, likely due to their background correlations with openness. 

## 5. Limitations

Our work has clear limitations. Even though the factor structure of our rumination-related variables was replicated in two samples, both samples consisted of college students, who may be atypical in many respects (e.g., a higher incidence of dysphoria and anxiety than the general population). We have no information on actual clinical diagnoses of depression or anxiety disorder for our participants. Longitudinal and/or experimental work would be necessary before causal conclusions can be derived. Finally, it remains to be seen if self-reported issues with letting go are indeed indicative of objective deficits in executive control.

## Figures and Tables

**Table 1 behavsci-12-00369-t001:** Results from factor analysis of the different rumination-related scales in both samples; principal component analysis with oblimin rotation; pattern matrix is presented.

	Sample 1			Sample 2			
	Factor 1 Repetitive Intrusive Thought	Factor 2 Reflectiveness	Factor 3 Inability to Let Go	Factor 4 Social Expressiveness	Factor 1 Repetitive Intrusive Thought	Factor 2 Reflectiveness	Factor 3 Inability to Let Go
RRQ Rumination	0.82				0.88		
WBSI Intrusion	0.82				0.84		
WBSI Suppression	0.72				0.78		
BRS Compulsivity	0.83				0.77		
BRS Worry	0.61				0.71		
BRS Distractibility	0.71				0.66		
BRS Broodiness	0.78				0.61		
BRS Optimism	−0.69				−0.47		
BRS Reflectiveness		0.86				0.84	
RRQ Reflection		0.86				0.80	
UBC-LG Social fear			0.89				0.89
UBC-LG Worry			0.86				0.86
UBC-LG Depression			0.94				0.84
UBC-LG Personal							0.58
BRS Social expressiveness				0.96		0.47	

Note. *n* = 423 for Sample 1 and 329 for Sample 2. RRQ = Rumination-Reflection Questionnaire; WBSI = white Bear Suppression Inventory; BRS = Broad rumination Scale; UBC-LG = University of British Columbia Cognition Inventory Letting Go scale. For legibility reasons, factor loadings below 0.40 are not represented.

**Table 2 behavsci-12-00369-t002:** Correlations between variables of interest for the regression analyses in Sample 1.

	1	2	3	4	5	6	7	8	9	10
1. Gender	1									
2. IPIP extraversion	−0.01	1								
3. IPIP agreeableness	0.18 ***	0.22 ***	1							
4. IPIP conscientiousness	0.03	−0.03	0.04	1						
5. IPIP neuroticism	0.34 ***	−0.01	0.12 *	−0.1 ***	1					
6. IPIP openness	0.04	0.06	0.2 ***	−0.04	−0.01	1				
7. Repetitive intrusive thought	0.25 ***	−0.08	0.14 **	−0.15 **	0.62 ***	0.02	1			
8. Reflectiveness	0.19 ***	0.01	0.29 ***	−0.01	0.19 ***	0.3 ***	0.26 ***	1		
9. Inability to let go	−0.01	−0.12 *	0.03	−0.01	0.05	0.01	0.08	−0.06	1	
10. STAI	0.20 ***	−0.08	0.08	−0.22 ***	0.55 ***	−0.05	0.54 ***	0.08	0.13 **	1
11. CES−D	0.19 ***	−0.05	0.09	−0.33 ***	0.53 ***	0.01	0.58 ***	0.19 ***	0.18 ***	0.60 ***

Note. *n* = 423. IPIP = International Personality Item Pool; STAI = State-Trait Anxiety Inventory; CES-D = Center for Epidemiological Studies-Depression. * *p* < 0.05. ** *p* < 0.01. *** *p* < 0.001.

**Table 3 behavsci-12-00369-t003:** Correlations between variables of interest for the regression analyses in Sample 2.

	1	2	3	4	5	6	7	8	9	10	11	12	13	14	15	16	17	18
1. Gender	1																	
2. IPIP extraversion	0.11	1																
3. IPIP agreeableness	0.31 ***	0.34 ***	1															
4. IPIP conscientiousness	0.06	−0.06	0.09	1														
5. IPIP neuroticism	0.24 ***	−0.01	0.03	−0.1	1													
6. IPIP openness	0.09	0.12 *	0.30 ***	0.10	−0.02	1												
7. FFMQ observing	0.11 *	0.09	0.18 **	0.07	0.07	0.24 ***	1											
8. FFMQ describing	0.01	0.37 ***	0.29 ***	0.21 ***	−0.18 **	0.26 ***	0.14 *	1										
9. FFMQ non−judging	0.12 *	0.09	0.04	0.26 ***	−0.43***	0.01	−0.26 ***	0.19 ***	1									
10. FFMQ acting with awareness	−0.01	0.03	0.08	0.36 ***	−0.35 ***	0.17 **	−0.11	0.29 ***	0.41 ***	1								
11. FFMQ nonreactivity	−0.24 ***	−0.08	−0.12 *	0.10	−0.54 ***	0.05	0.22 ***	0.14 *	0.14 *	0.16 **	1							
12. Self−compassion	−0.05	0.17 **	0.13 *	0.16 **	−0.52 ***	0.16 **	0.23 ***	0.36 ***	0.34 ***	0.29 ***	0.52 ***	1						
13. Self−transcendence	0.10	0.35 ***	0.45 ***	0.21 ***	−0.35 ***	0.27 ***	0.24 ***	0.42 ***	0.28 ***	0.25 ***	0.26 ***	0.53***	1					
14. Repetitive intrusive thought	0.17 **	−0.11 *	0.1	−0.22 ***	0.63 ***	0.02	0.16 **	−0.25 ***	−0.54 ***	−0.50 ***	−0.42 ***	−0.49 ***	−0.32 ***	1				
15. Reflectiveness	0.11 *	0.15 **	0.33 ***	0.06	0.03	0.48 ***	0.33 ***	0.32 ***	−0.07	0.05	0.07	0.32 ***	0.35 ***	0.13 *	1			
16. Inability to let go	0.07	−0.16 **	0.01	−0.08	0.35 ***	−0.11 *	−0.02	−0.15 **	−0.32 ***	−0.29 ***	−0.21 ***	−0.31 ***	−0.26 ***	0.47 ***	0.05	1		
17. CES−D	0.06	−0.03	0.04	−0.30 ***	0.43 ***	0.01	0.10	−0.19 ***	−0.40 ***	−0.43 ***	−0.24 ***	−0.36 ***	−0.38 ***	0.57 ***	0.13 *	0.54 ***	1	
180. STAI	0.03	−0.07	−0.06	−0.26 ***	0.53 ***	−0.10	0.00	−0.36 ***	−0.39 ***	−0.37 ***	−0.40 ***	−0.48 ***	−0.49 ***	0.59 ***	−0.02	0.42 ***	0.59 ***	1
19. Psychological wellbeing	0.11 *	0.40 ***	0.39 ***	0.30 ***	−0.32 ***	0.26 ***	0.15 **	0.47 ***	0.30 ***	0.34 ***	0.25 ***	0.46 ***	0.74 ***	−0.37 ***	0.38 ***	−0.32 ***	−0.43 ***	−0.50 ***

Note. *n* = 329. IPIP = International Personality Item Pool; STAI = State-Trait Anxiety Inventory; CES-D = Center for Epidemiological Studies-Depression. * *p* < 0.05. ** *p* < 0.01. *** *p* < 0.001.

**Table 4 behavsci-12-00369-t004:** Results from multiple regression analyses predicting dysphoria (CES-D), anxiety (STAI), and psychological well-being (PWBS) from gender, the Big Five, repetitive intrusive thought, reflectiveness, inability to let go, and mood. All regression coefficients are beta coefficients (i.e., standardized coefficients).

	CES−D		STAI		PWBS
	Sample 1	Sample 2	Sample 1	Sample 2	Sample 2
**Step 1**					
Gender	0.02	−0.04	0.03	−0.07	0.05
IPIP extraversion	−0.06	−0.05	−0.11	−0.04	0.30 ***
IPIP agreeableness	0.07	0.06	0.07	−0.04	0.24 ***
IPIP conscientiousness	−0.26 ***	−0.28 ***	−0.15 ***	−0.20 ***	0.26 ***
IPIP neuroticism	0.48 ***	0.41 ***	0.50 ***	0.52 ***	−0.30 *
IPIP openness	−0.01	0.02	−0.04	−0.05	0.12 ***
*R*^2^	0.35 ***	0.26 ***	0.32 ***	0.33 ***	0.43 ***
**Step 2**					
Gender	0.00	−0.04	0.02	−0.08	0.06
IPIP extraversion	−0.01	0.06	−0.07	0.05	0.23 ***
IPIP agreeableness	0.01	−0.04	0.05	−0.09	0.22 ***
IPIP conscientiousness	−0.24 ***	−0.21 ***	−0.13 **	−0.13 **	0.20 ***
IPIP neuroticism	0.23 ***	0.11	0.30 ***	0.25 ***	−0.14 **
IPIP openness	−0.04	0.01	−0.03	−0.03	−0.02
Repetitive intrusive thought	0.38 ***	0.29 ***	0.33 ***	0.35 ***	−0.21 ***
Reflectiveness	0.04	0.08	−0.06	−0.02	0.30 ***
Inability to let go	0.13 **	0.35 ***	0.08 *	0.18 **	−0.16 ***
*R*^2^	0.45 ***	0.47 ***	0.39 ***	0.45 ***	0.53 ***
*R*^2^ change	0.10 ***	0.21 ***	0.07 ***	0.12 ***	0.11 ***
**Step 3**					
Gender	−0.01	−0.02	0.02	−0.06	0.04
IPIP extraversion	0.02	0.04	−0.07	0.03	0.26 ***
IPIP agreeableness	0.00	−0.01	0.04	−0.08	0.19***
IPIP conscientiousness	−0.20 ***	−0.18 ***	−0.04	−0.06	0.13 **
IPIP neuroticism	0.14	0.04	0.22 ***	0.21 ***	−0.07
IPIP openness	−0.03	0.02	−0.02	−0.03	−0.02
Repetitive intrusive thought	0.27 ***	0.18 **	0.19 **	0.26 ***	−0.08
Reflectiveness	0.06	0.09	−0.08	−0.05	0.32 ***
Inability to let go	0.10 **	0.30 ***	0.04	0.07	−0.04
CES−D	(NA)	(NA)	0.36 ***	0.31 ***	−0.22 ***
STAI	0.32 ***	0.30 ***	(NA)	(NA)	−0.20 ***
*R*^2^	0.52 ***	0.52 ***	0.46 ***	0.50 ***	0.77 ***
*R*^2^ change	0.06 ***	0.05 ***	0.07 ***	0.04 ***	0.06 ***

Note. *n* = 423 for Sample 1 and 329 for Sample 2. IPIP = International Personality Item Pool; STAI = State-Trait Anxiety Inventory; CES-D = Center for Epidemiological Studies-Depression; PWB = Psychological Well Being Scale. * *p* < 0.05. ** *p* < 0.01. *** *p* < 0.001. Mediation analyses.

**Table 5 behavsci-12-00369-t005:** Results from Sobel tests examining whether the inability to let go, repetitive intrusive thought, and reflectiveness significantly mediate between the seven mindfulness variables (FFMQ, self-compassion, and self-transcendence) and the three outcomes of dysphoria (CES-D), anxiety (STAI), and psychological well-being in Sample 2. All regression coefficients are beta coefficients (i.e., standardized coefficients).

	CES-D			STAI			Psychological Well-Being		
	Repetitive Intrusive Thought	Reflectiveness	Inability to Let Go	Repetitive Intrusive Thought	Reflectiveness	Inability to Let Go	Repetitive Intrusive Thought	Reflectiveness	Inability to Let Go
FFMQ observing	2.91 **	2.19 *	0.34	2.92 **	0.38	0.35	2.77 **	4.84 ***	0.35
FFMQ describing	4.40 ***	2.18 *	2.63 **	4.44 ***	0.37	2.57 *	3.94 ***	4.72 ***	2.48 *
FMQ non-judging	8.41 ***	1.08	5.31 ***	8.71 ***	0.36	4.88 ***	6.11 ***	1.2	4.31 ***
FFMQ acting with awareness	7.97 ***	0.84	4.95 ***	8.22 ***	0.35	4.60***	5.93 ***	0.9	4.11 ***
FFMQ nonreactivity	6.93 ***	1.14	3.59 ***	7.10 ***	0.36	3.45 ***	5.47 ***	1.28	3.23 **
Self-compassion	7.84 ***	2.18 *	5.14 ***	8.09 ***	0.38	4.74 ***	5.88 ***	4.71 ***	4.22 ***
Self-transcendence	5.53 ***	2.21 *	4.33 ***	5.62 ***	0.38	4.17 ***	4.70 ***	4.99 ***	3.80 ***

Note. *n* = 329. FFMQ = Five Facets Mindfulness Questionnaire; STAI = State-Trait Anxiety Inventory; CES-D = Center for Epidemiological Studies-Depression. * *p* < 0.05. ** *p* < 0.01. *** *p* < 0.001.

**Table 6 behavsci-12-00369-t006:** Results from multiple regression analyses predicting the inability to let go, repetitive intrusive thought, and reflectiveness from gender, the Big Five, and the mindfulness variables in Sample 2. All regression coefficients are beta coefficients (i.e., standardized coefficients).

	Inability to Let Go	Repetitive Intrusive Thought	Reflectiveness
Gender	0.04	0.05	0.02
IPIP extraversion	−0.15 *	−0.10 *	−0.07
IPIP agreeableness	0.12 *	0.14 **	0.10
IPIP conscientiousness	0.05	−0.04	−0.03
IPIP neuroticism	0.15 *	0.26 ***	0.15 *
IPIP openness	−0.07	0.07	0.33 ***
FFMQ observing	−0.07	0.11 **	0.09
FFMQ describing	0.05	−0.03	0.13
FFMQ non−judging	−0.17 *	−0.20 ***	−0.11
FFMQ acting with awareness	−0.16 **	−0.23 ***	−0.04
FFMQ nonreactivity	−0.00	−0.13 **	−0.05
Self−compassion	−0.06	−0.13 *	0.29 ***
Self−transcendence	−0.09	−0.07	0.12
*R* ^2^	0.22 ***	0.60 ***	0.39 ***

Note. *n* = 329. IPIP = International Personality Item Pool; FFMQ = Five Facets Mindfulness Questionnaire. * *p* < 0.05. ** *p* < 0.01. *** *p* < 0.001.

**Table 7 behavsci-12-00369-t007:** Results from multiple regression analyses predicting dysphoria (CES-D), anxiety (STAI), and psychological well-being (PWB) from gender, the Big Five, the mindfulness variables, repetitive intrusive thought, reflectiveness, and the inability to let go in Sample 2. All regression coefficients are beta coefficients (i.e., standardized coefficients).

	CES−D	STAI	Psychological Well−Being
Gender	−0.03	−0.08	0.06
IPIP extraversion	0.11 *	0.13 **	0.15 ***
IPIP agreeableness	0.07	0.04	0.05
IPIP conscientiousness	−0.16 **	−0.07	0.14 ***
IPIP neuroticism	0.05	0.15 *	−0.01
IPIP openness	0.02	−0.01	−0.03
FFMQ observing	0.07	0.06	−0.06
FFMQ describing	−0.03	−0.18 ***	0.08 *
FFMQ non−judging	0.01	0.03	−0.01
FFMQ acting with awareness	−0.11 *	−0.01	0.06
FFMQ nonreactivity	0.04	−0.09	0.10 *
Self−compassion	−0.06	−0.04	−0.09
Self−transcendence	−0.27 ***	−0.30 ***	0.51 ***
Repetitive intrusive thought	0.17 *	0.22 **	−0.08
Reflectiveness	−0.15 **	0.09	0.20 ***
Inability to let go	0.33 ***	0.15 **	−0.11 **
*R* ^2^	0.53 ***	0.53 ***	0.67 ***

Note. *n* = 329. IPIP = International Personality Item Pool; FFMQ = Five Facets Mindfulness Questionnaire; STAI = State-Trait Anxiety Inventory; CES-D = Center for Epidemiological Studies-Depression. * *p* < 0.05. ** *p* < 0.01. *** *p* < 0.001.

## Data Availability

Data can be accessed at https://osf.io/hr85n/.

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
