# Peer review of "Letting Go as an Aspect of Rumination and Its Relationship to Mindfulness, Dysphoria, Anxiety, and Eudemonic Well-Being"

_behavsci, 2022, doi:10.3390/bs12100369_

Round 1

Reviewer 1 Report

Review of the article "Letting go as an aspect of rumination and its relationship to mindfulness, dysphoria, anxiety, and eudemonic well-being”

Presented in a review article, "Letting go as an aspect of rumination and its relationship to mindfulness, dysphoria, anxiety, and eudemonic well-being" is devoted to the description of the results of original research conducted to an aspect of rumination and its relationship to mindfulness, dysphoria, anxiety, and eudemonic well-being. According to the data obtained, the finding suggest that letting go is a potentially interesting aspect of rumination not fully captured in the traditional concept of rumination and its standard measures.

1. Overall assessment of the work

1. The article title and abstract are appropriate. The purpose of the article and its significance is stated clearly. The content of the manuscript corresponds to its title, as well as the profile of the journal as a whole.

2. The style of presentation of the material is characterized by literary and scientific literacy of the text, terminological clarity and legibility for a potential professional audience.

3. The structure of the article corresponds to the APA-style: there is an introduction to the state of the problem, the objectives of the study are described, the research program (sample, methods and stages of the study), the results of the study and their description, discussion of the results, conclusion are described in detail.

4. The information presented in the review part of the manuscript (introduction, description of the relevance and objectives of the research, as well as discussion of the results) is relevant: the text and the list of references contain references to modern (not older than 5 years) research in this field, analogues and prototypes of the conducted research, problem reviews in leading specialized peer-reviewed scientific journals.

5.The study methods are sound and appropriate, the writing is clear and concise.

The conclusions or summary are accurate and supported by the content. The article is of interest to members of the education research community

6. The research material and methods correspond to the set goals and objectives. The results obtained were checked for the normality of the variation series, the criteria of statistical significance and the level of reliability were adequately selected, and the methods of multidimensional statistics were used. The structure and size of the surveyed sample allowed us to obtain statistically reliable results.

7. The conducted study complies with the norms of medical ethics and the GCP (Good Clinical Practice) standard; the study protocol was approved by the Local ethics committee, all participants signed an informed consent form.

8. The recommendations and conclusions of the study can be used in the practical work of clinical psychologists and tutors.

9. Graphic materials (figures, tables, etc.) are presented in sufficient quantity and readable.

2. Comments and recommendations. There are no fundamental comments.

3. Final Conclusion. I believe that the article "Letting go as an aspect of rumination and its relationship to mindfulness, dysphoria, anxiety, and eudemonic well-being" has significant scientific and practical significance. The article meets all the requirements for scientific articles, as well as international standards of bibliographic and abstract databases and can be published in the Journal “Behavioral Sciences”.

Author Response

Thank you. There did not seem to be actionable items in this review, so no changes in the paper were made in response to this review.

Reviewer 2 Report

1. The title of the article is adequate to its content. The abstract is well-written.

2. The introduction concisely and clearly presents the analysed variables and their interconnectedness.  However, the introduction should be completed with the indication of relations between adaptive and maladaptive rumination and personality variables in the view of Big Five theory (as they are the subject of the analyses undertaken in the following parts of the text).

3. The description of participants is synthetic but it includes the necessary information and is therefore sufficient.

4. Multiple research tools used in the study were clearly presented. It would be a good idea to supplement the description with the reliability indicators of these tools.

5. The results are clearly presented and well-organised.

6. In my opinion Table 4 should be moved from Discussion to Results. Apart from that, the discussion is well-organised , clear and comprehensive.

Author Response

We have now added the following paragraph to the introduction:

Note that we included a measure of personality, notably the Big Five, as a control measure, to partial out any influence of personality as a background measure. Such background or third-variable relationships are to be expected. For instance, neuroticism has often been found to correlate with both rumination and depression (e.g.., Conway, Csank, Holm, & Blake, 2000; Wupperman & Neumann, 2006) as well as with mindfulness (Giluk, 2009); consequently, correlations between these three variables might well be due solely or in part to their underlying relationship with this personality trait. Likewise, conscientiousness is positively related to mindfulness (Giluk, 2009) and negatively related to anxiety and depression (e.g., Akram, Gardani, Akram, & Allen, 2019) as well as rumination (e.g., Lyon, Elliott, Brown, Eszlari, & Juhasz, 2020), creating the potential of background correlations between these variables as well.

Table 4 should, of course, be placed between Table 3 and Table 5 in Results, and not in the Discussion section. We  delivered our paper to the journal in APA-style, that is, with tables at the end of the manuscript, so this was not our doing. We have now rectified this. 

We have added Cronbach alpha values for all of our scales in the manuscript.

Reviewer 3 Report

The article focuses on the experimental demonstration of the existence, hitherto neglected by the literature, of the “inability to let go” as an aspect of rumination, also highlighting its role of moderation with respect to other psychological outcomes (dysphoria, anxiety, and well-being). The experimental design appears robust and coherent, supported by a valid analysis of the literature on rumination and by logically founded and original conclusions. Nevertheless, I would like to suggest two minor revisions, which in my opinion would further elevate its (already high) overall quality:

1) In literature review a little space is dedicated to the deepening of the relationship between psychological well-being and rumination, as well as to a further definition of the concept of well-being. Furthermore, it is not clear why eudaimonic well-being (in the text "eudemonic", but I think the term reported here is more correct) was chosen, and not the hedonic one. I would invite the authors to further explore these points.

2) The presence of a factorial discrepancy between the two experimental groups, where in Group 1 there is a fourth factor strongly associated with the "BRS Social expressiveness" absent in group 2, does not appear to be adequately problematized. I would invite the authors to further interpret this result.

Author Response

We have now added the following paragraph to the introduction: 

Well-being is often operationalized as either hedonic (i.e., with a focus on pleasure or happiness—feeling good) or eudemonic (i.e., flourishing or actualization) (Ryan & Deci, 2001). We chose the latter as our outcome measure because it is more multidimensional and less directly related to our other outcomes, dysphoria and anxiety; eudemonic wellbeing is also more stable over time and has been found to be the driver of hedonic well-being rather than the other way around (Joshanloo, 2019). Previous work (e.g., Harrington & Loffredo, 2011) has shown a modest connection between reflection and eudemonic well-being and a stronger connection with maladaptive rumination; the question whether letting-go would be a predictor is still open.  

We have now added the following footnote to the Discussion:

Note that there was a discrepancy in the factor analyses, such that only the first three factors were identical across the two samples; Sample 1 additionally yielded a fourth factor consisting of a single scale (Social Expressiveness). Given that both samples are drawn from the same population, given that single-item factors are hard to interpret, given that Social Expressiveness per se does not function in our theoretical framework, and given that Social Expressiveness did not load on any factor in Sample 2, we decided to move forward with the three factors both samples had in common.   

Reviewer 4 Report

The topic is approppriate for this special issue.

Introduction

The paper is well written, includes relevant references and presents all the concepts used in the research. 

I would recommend to include a short justification for using Mini-International Personality Item Pool.

Results

For a better understanding I would add some graphical representation besides the tables that include all the results - probably in form of a summary of the most important outcomes.

Generally, the paper structure could be slighlty improved - by including numbering, some paragraph summarizing the next ideas, ...

Finally, I appreciate the hard work and the quality of this research.

Author Response

We have now added the following justification to the Introduction:

Note that we included a measure of personality, notably the Big Five, as a control measure, to partial out any influence of personality as a background measure. Such background or third-variable relationships are to be expected. For instance, neuroticism has often been found to correlate with both rumination and depression (e.g.., Conway, Csank, Holm, & Blake, 2000; Wupperman & Neumann, 2006) as well as with mindfulness (Giluk, 2009); consequently, correlations between these three variables might well be due solely or in part to their underlying relationship with this personality trait. Likewise, conscientiousness is positively related to mindfulness (Giluk, 2009) and negatively related to anxiety and depression (e.g., Akram, Gardani, Akram, & Allen, 2019) as well as rumination (e.g., Lyon, Elliott, Brown, Eszlari, & Juhasz, 2020), creating the potential of background correlations between these variables as well.

The results are, we feel, quite simple. As we summed up at the beginning of the Discussion:

The two main results from the study are (a) that letting go can indeed plausibly be considered an aspect of rumination not fully captured in the standard way of measuring the concept, and (b) that the ability to let go indeed mediates between mindfulness and the psychological outcomes considered here.

It seems to us that making a graphical representation of these findings would not be much more straightforward than the verbal expression above.

We believe the journal does not allow numbering of sections. We are also somewhat puzzled at the comment about structuring, because we make ample use of paragraph headers and enumeration (first, second, ...) throughout to aid the reader. We would appreciate more direct instruction as to where structural elements were missing or could be useful.